# Distinguishing Ignorance from Error in LLM Hallucinations

## Abstract

Large language models (LLMs) are susceptible to hallucinations—outputs that are ungrounded, factually incorrect, or inconsistent with prior generations. We focus on close-book Question Answering (CBQA), where previous work has not fully addressed the distinction between two possible kinds of hallucinations, namely, whether the model (1) does not hold the correct answer in its parameters or (2) answers incorrectly despite having the required knowledge. We argue that distinguishing these cases is crucial for detecting and mitigating hallucinations. Specifically, case (2) may be mitigated by intervening in the model's internal computation, as the knowledge resides within the model's parameters. In contrast, in case (1) there is no parametric knowledge to leverage for mitigation, so it should be addressed by resorting to an external knowledge source or abstaining. To help distinguish between the two cases, we introduce **W**rong **A**nswer despite having **C**orrect **K**nowledge (WACK), an approach for constructing model-specific datasets for the second hallucination type. Our probing experiments indicate that the two kinds of hallucinations are represented differently in the model's inner states. Next, we show that datasets constructed using WACK exhibit variations across models, demonstrating that even when models share knowledge of certain facts, they still vary in the specific examples that lead to hallucinations. Finally, we show that training a classifier on our WACK datasets leads to better hallucination detection of case (2) hallucinations than using the common generic one-size-fits-all datasets.

## 1 Introduction

Large Language Models (LLMs) are prone to generating outputs that lack grounded in the model's input or real-world facts, as well as outputs that may be inconsistent with earlier generations within the same session (Ji et al., 2023; Sharma et al., 2023; Kalai & Vempala, 2023). These issues, collectively known as *hallucinations*, are critical to address due to their impact on LLM reliability.

Numerous studies have focused on the detection and mitigation of hallucinations (e.g. Li et al. (2023); Zhang et al. (2024); Marks & Tegmark (2023); Chen et al. (2024); CH-Wang et al. (2023)). However, existing work often fails to distinguish between the different causes of hallucinations, conflating two distinct types: first type, denoted as $\mathsf{HK}^-$, refers to cases where the model lacks the required information, leading it to hallucinate. The second, denoted as $\mathsf{HK}^+$, type occurs when, although the model has the necessary knowledge and can generate correct answers under certain prompts, it still produces an incorrect response in a different but similar prompt setting. These types represent fundamentally different problems, requiring different solutions: When a model lacks knowledge one should consult external sources (or abstain), but when a model has the knowledge it may be possible to intervene in its computation to obtain the correct answer. Failing to differentiate between these causes can weaken the effectiveness of detection and mitigation techniques, which often categorize outputs simply as either 'hallucination' or 'factually correct' without further investigating these two distinct types of hallucination (Marks & Tegmark, 2023; Azaria & Mitchell, 2023; Li et al., 2023; Rateike et al., 2023; Zhang et al., 2024; Chen et al., 2024; Zou et al., 2023; Hoscilowicz et al., 2024).

Our first contribution in this work is an automatic approach to obtain **W**rong **A**nswers despite having **C**orrect **K**nowledge (WACK). This approach constructs a *model-specific* hallucination dataset that captures the distinction between the two types of hallucinations. It differentiates between hallucinations caused by a lack of knowledge ($\mathsf{HK}^-$) and those caused by incorrect generation despite

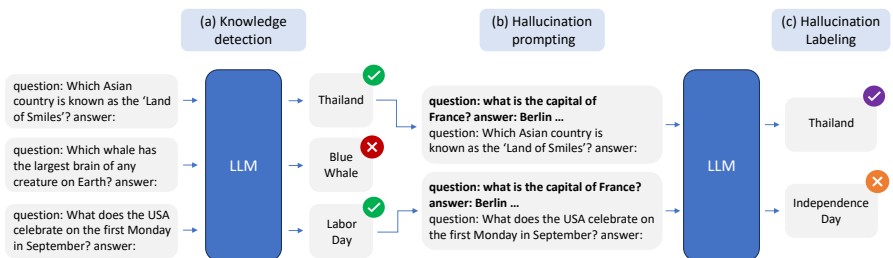

Figure 1: **WACK Setup:** (a) The first step in our process involves detecting whether the model knows the correct answer. If the model does not know the correct answer, the example is labeled as hallucination caused by not knowing (HK$^-$) If the model knows the correct answer, we proceed to the next stage. (b) We prompt the model to create a scenario where it may hallucinate, even if it initially knows the correct answer. Here we show a snowballing bad-shots prompt. (c) Under the new setting, if the model generates the correct answer, the example is labeled as factually-correct; otherwise, it is labeled as a hallucination despite knowing (HK$^+$).

the existence of the knowledge in the target model (HK$^+$). The process infers by automatically categorizing examples based on the model's knowledge via inspection of correct responses in output samples. If the model lacks the required knowledge, the example is labeled as HK$^-$. Otherwise, for examples where the model possesses the correct knowledge, it further splits the cases into "factually-correct" and HK$^+$ based on the model generation in an altered prompt setting. This novel alternate prompt setting employs persuasion (Xu et al., 2023; Zeng et al., 2024), weak semantics (Yao et al., 2023) that aim to modify the prompt to reduce its semantic content, and other techniques to induce hallucinations in scenarios that mimic regular interactions with an LLM. See the WACK methodin Figure 1.

We construct WACK datasets for three state-of-the-art LLMs of the 7B–9B size range. With these tailored benchmarks, we investigate how different types of hallucinations (HK$^-$ and HK$^+$) are represented within the models. We do so by training classifiers on the model's inner states, a common method in the field of hallucination detection (CH-Wang et al., 2023; Azaria & Mitchell, 2023). While prior work detects whether any hallucination occurs, we show it is possible to distinguish between the two hallucination types (Section 4.1). Next, we focus on HK$^+$ type of hallucinations and show the generalization of WACK between different prompt settings; a classifier trained on examples from one setting is able to predict hallucinations in another setting (Section 4.2). Lastly, we show that WACK datasets differ across models, highlighting the significance of using model-specific datasets that account for each model's unique knowledge and hallucination patterns (Section 5.1). As a result, we demonstrate that model-specific datasets are more effective for HK$^+$ detection than generic datasets (Sections 5.2 and 5.3).

Our main contributions are as follows: (I) We propose WACK, a methodology for constructing a model-specific dataset that includes factually correct and hallucination examples both due to lack of knowledge (HK$^-$) and despite having knowledge (HK$^+$). We will release the datasets for the models we experimented with. (II) We demonstrate that a model's internal states can be used to distinguish between the two hallucination types. (III) We demonstrate the importance of model-specific datasets for HK$^+$ hallucination detection.

## 2 MODEL-SPECIFIC DATASET CONSTRUCTION

In this section, we outline the process of creating a model-specific dataset, with a focus on generating HK$^+$ hallucination examples. Figure 1 provides a detailed overview of this setup. The process begins by classifying examples based on the model's knowledge, labeling all instances where the model lacks knowledge as HK$^-$. Next, using examples where the model knows the correct answer, we create a scenario in which hallucinations can occur despite the model's knowledge (HK$^+$). The next two subsections describe these steps. We focus on closed-book question answering (CBQA) tasks with short answers.

## 2.1 CATEGORIZATION OF KNOWLEDGE

In our CBQA setting, a model is given a question $q$ and generates an answer $\tilde{a}$, which may match the factually correct gold answer $a_g$ or else constitute a hallucination. Knowledge in a language model can be viewed as lying on a spectrum. We refer to the model's parametric knowledge as being at the 'low-knowledge end' when there is little to no association between $a_g$ and $q$, and as the 'high-knowledge end' when this association is strong. Hallucinations at the low-knowledge end of the spectrum are somewhat expected, as the model is unlikely to generate $a_g$ (that is, we expect $\tilde{a} \neq a_g$). However, hallucinations can occur anywhere along this spectrum, including at the high-knowledge end. Detecting the cause of hallucinations in the middle of the spectrum is more complex, as they may arise either from insufficient knowledge or despite adequate knowledge.

To simplify our analysis, we focus on the two ends of the spectrum: high-knowledge and low-knowledge, which still provide a compelling overview of the two types of hallucinations. To this end, given model $M$, we follow the setup of Gekhman et al. (2024) in which $M$ generates various completions to $q$, and then we verify the existence of the answer $a_g$ in the output. Specifically[1], we perform one greedy generation plus five generations with a temperature of $0.5$. We use a 3-shots in-context learning scenario (Brown et al., 2020), generate a maximum of 5 tokens, and look for an exact match to $a_g$. If the model did not generate $a_g$ in any of the generations, the example is labeled $\mathsf{HK}^-$. If the model generates $a_g$ in all the attempts, this example is considered a high-knowledge scenario and we next label it as either factually-correct or $\mathsf{HK}^+$.

## 2.2 HALLUCINATION DESPITE KNOWLEDGE

To label a high-knowledge example as either factually correct or $\mathsf{HK}^+$, we follow Zhang et al. (2023), who demonstrated that after a model produced an incorrect answer, it was likely to generate an incorrect explanation to justify its error, which they termed the "snowballing effect". Similar behaviors were also shown when using persuasion techniques that modify the prompt to include persuasions (Xu et al., 2023; Zeng et al., 2024), and weak semantics (Yao et al., 2023) that modify the prompt to reduce its semantic content.

We argue that these settings are important to focus on as they reflect mistakes in the prompt, which can originate from either the user or the model's previous generation.

1. User mistakes: We cannot expect users to have perfect knowledge—they can write a wrong fact or make grammar or language mistakes—and if their own errors cause the model to hallucinate, this represents a real-world problem that needs to be addressed.
2. Model's previous mistakes: The model may create a snowballing effect on its own by generating mistakes in previous turns that would be added to the prompt (e.g., Zhang et al. 2023). If the model produces a hallucination, we do not want this to affect subsequent generations, thus this issue should also be mitigated.

To facilitate the creation of $\mathsf{HK}^+$ hallucinations in large quantities, we design two synthetic setups that align with the ideas described above: (a) *Bad-shots*, leveraging snowballing, and (b) *Alice-Bob*, leveraging persuasion and weak semantics. Later in the paper, we demonstrate that one setting generalizes to the other one, indicating that it is valid to use these specific setups to investigate the general phenomenon of $\mathsf{HK}^+$.

**Bad-shots setting.** This setting illustrates how mistakes in the context can create a cascading effect that may compromise the accuracy of the model's subsequent generations. To imitate the snowballing effect on a large scale, we propose a synthetic method which we name *Bad-Shot Prompting*. We construct 20 false QA pairs using ChatGPT 3.5 (OpenAI, 2022), where the false answer is semantically similar to the correct one. For instance, here is a *good-shot* example and its corresponding *bad-shot* example:

> Good-shot  question: Which element has the chemical symbol 'H'? answer: Hydrogen
> Bad-shot  question: Which element has the chemical symbol 'H'? answer: Helium

---

[1]The decision process of the following hyperparams can be found in Appendix A.

> **question:** In what year did World War II end?
> **answer:** 1939
> **question:** What is the smallest prime number?
> **answer:** 1
> **question:** Who wrote 'Romeo and Juliet'?
> **answer:** Jane Austen
> **question:** What does the USA celebrate on the first Monday in September?
> **answer:**

Figure 2: Example prompt using 3-bad-shot snowballing. Depending on the model's response, this example would be labeled as either 'factually correct' or 'hallucinated' (HK$^+$).

Table 1: Generated answers using greedy decoding on TriviaQA.

| Model | Prompt | Generation | |
|---|---|---|---|
| | | w/ good-shots | w/ bad-shots |
| Gemma-2-9B | question: What does the USA celebrate on the first Monday in September? answer: | Labor day | Independence Day |
| Llama-3.1-8B | question: In which Disney film do the fairies Flora, Fauna and Merryweather appear? answer: | Sleeping Beauty | The Lion King |
| Mistral-7B-v0.3 | question: What is the official spoken language of China? answer: | Mandarin | English |

For each target question in the eval set we prepend bad-shot examples in a few-shot manner, thus simulating mistakes that a user or a model might create as part of the input context. In practice, we use 3 random bad-shots before each example from the dataset, as demonstrated in Figure 2.

As we will see below, the bad-shot setting enables us to obtain many examples in which models hallucinate despite having the knowledge to answer correcly (HK$^+$). In Table 1 we show one example per model from the TriviaQA dataset (Joshi et al., 2017), where the model generated the correct answer using 3 random good shots (regular-few-shots), but hallucinated when prompted with 3 random bad shots.

**Alice-Bob setting.** The Alice-Bob setting uses persuasion and weak semantics in the form instead of snowballing bad shots. In this setting, we add the following text at the beginning of the prompt, along with a one-shot example at the end:

> Alice and Bob are students studying for a test on general world knowledge that requires a vast amount of understanding and knowledge. Alice is the smarter of the two. They are given four questions in total. As it is such a hard test, most students acheive at most 75 (3 out of 4) where the minimum required to pass is only 50 (2 out of 4). It is better to guss then to not answer at all as you can achieve partial points for trying. Bob and Alice only need to pass the test. Alice will ask the questions and Bob will try his best.

This setting is more subtle than the bad-shots setting and generates fewer hallucinations. Drawing on ideas from related studies (Xu et al., 2023; Yao et al., 2023; Zeng et al., 2024), the aim is to simulate a persuasion scenario with a few deliberate mistakes (underlined in the prompt). The prompt typos are intended to mimic small inaccuracies that simulate an error. The persuasive aspect of the setting comes through several nuances in the text: (1) there is an implication that Bob is not smart, (2) the test is described as difficult, (3) to pass, one only needs to be correct on 2 out of the 4 questions, and (4) there is no suggestion that exceeding the minimum required score offers any advantage.

Table 2: Dataset labels statistics using bad-shot setting.

| Dataset | # Factually correct | # Hallucination (HK$^+$) | # Do-not-know (HK$^-$) |
|---|---|---|---|
| TriviaQA-Llama3-WACK | 14154 | 1675 | 7356 |
| Natural-Questions-Llama3-WACK | 5934 | 1104 | 14739 |
| TriviaQA-Gemma-WACK | 13534 | 2563 | 6991 |
| Natural-Questions-Gemma-WACK | 6045 | 1859 | 13762 |
| TriviaQA-Mistral-WACK | 12652 | 2841 | 7650 |
| Natural-Questions-Msitral-WACK | 5562 | 1546 | 14689 |

## 2.3 DATASET CONSTRUCTION

Equipped with our process for separating examples of low and high knowledge (Section 2.1) and and further labeling high-knowledge examples (Section 2.2), we create model-specific datasets. As sources for examples to label, we use two common closed-book question answering datasets: TriviaQA (Joshi et al., 2017) and NaturalQuetions (Kwiatkowski et al., 2019). We experiment with three models: Mistral-7B-v0.3 (Jiang et al., 2023), Llama-3.1-8B (Dubey et al., 2024) and Gemma-2-9B (Team et al., 2024).

Table 2 provides the number of examples in each category for the resulting model-specific datasets. We observe that even under the bad-shots setting, most of the model's high-knowledge examples are labeled as factually correct rather than hallucinations. Still, we are left with sufficient cases of hallucinations-despite-knowledge (HK$^+$), which we use in the subsequent sections. In the Alice-Bob setting, we observe similar trends, but with fewer hallucinations (Appendix C). For more details regarding the dataset construction, see Appendix B.

## 3 IMPLEMENTATION DETAILS

We aim to show the importance of separating the two hallucination types and using our model's specific dataset to create better detectors. In the following sections, we report on various experiments for detecting different types of hallucinations by training classifiers on inner model states. In all detection experiments, we randomly select 1000 examples from each label for analysis in each dataset and split them to 70%/30% for training/test.[2] We use a linear classifier for detection, as in prior work (Li et al., 2023; CH-Wang et al., 2023).[3] The detection results in the main paper are on hidden states from the residual component (after each Transformer block); see Appendix D for similar results on the MLP and Attention components. Each experiment was repeated with three random seeds for the SVM and split into training/test sets. We report average results with standard deviations. To maintain consistency with the prompts used in the creation of the WACK dataset, all examples incorporate similar prompts (bad shots or Alice-Bob). In addition, unless stated otherwise the results are shown under the bad-shot setting and the dashed black line is the baseline. Lastly, unless stated otherwise, given an example we detect at the answer ($\tilde{a}$) last token, which may or may not be a hallucination.

All experiments were run on NVIDIA RTX 6000 Ada (49GB) with 4 CPUs. Generating all the datasets and results takes approximately 2 weeks on one GPU.

## 4 DETECTING AND MITIGATING DIFFERENT TYPES OF HALLUCINATIONS

We first investigate whether different types of hallucinations are represented differently inside models. Then we examine mitigation and generalization between settings.

### 4.1 MODELS CAN DISTINGUISH HK$^+$ FROM HK$^-$ FROM FACTUALLY CORRECT

We first explore the distinction between hallucinations arising from a model's lack of knowledge and those that occur even when the model possesses relevant information. This differentiation is

---

[2]In cases where there are fewer than 1000 hallucinations we use all the hallucinations we have.

[3]We ran the detection on normalized vectors of the model's inner states at the last token using a linear SVM.

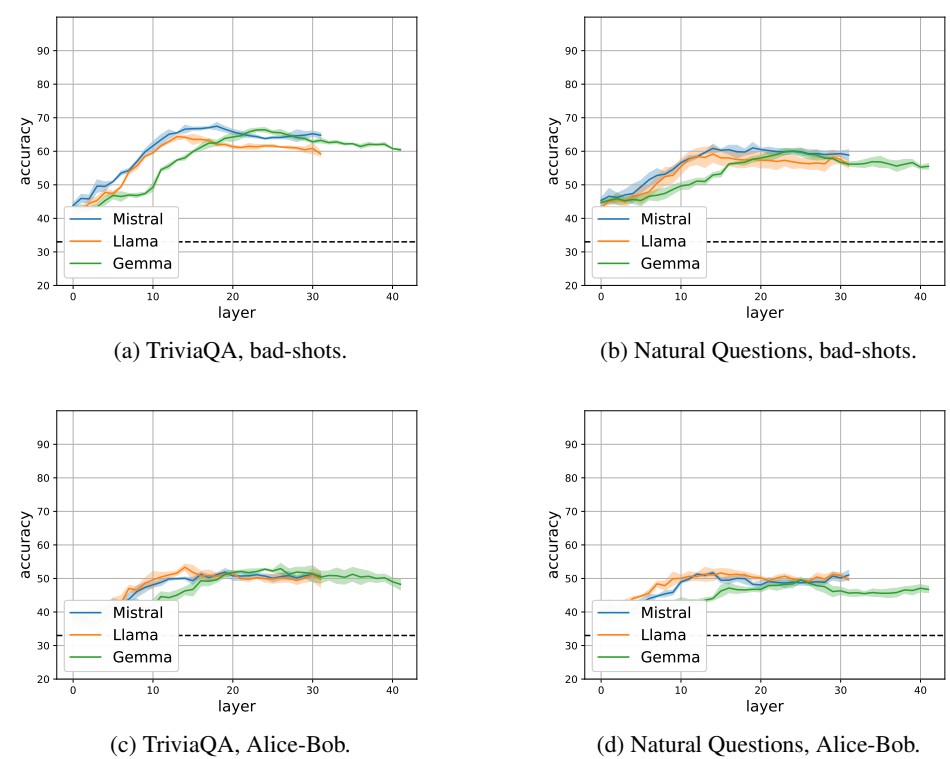

(a) TriviaQA, bad-shots.

(b) Natural Questions, bad-shots.

(c) TriviaQA, Alice-Bob.

(d) Natural Questions, Alice-Bob.

Figure 3: 3-way classification results into (i) hallucinations caused by lack of knowledge ($\mathsf{HK}^-$), (ii) hallucinations caused despite having knowledge ($\mathsf{HK}^+$), and (iii) factually correct examples. We show result using a bad-shot setting (Top) or an Alice-bob setting (Bottom).

crucial for understanding hallucinations' underlying mechanisms and developing targeted detection and mitigation strategies. We employ detection from the model's inner state to demonstrate that the model represents these hallucinations differently. Note that this is a challenging task, as it not only requires distinguishing hallucinations from factually correct responses, but also determining the type of hallucination. This involves an understanding of the model's knowledge, which must be considered.

To address a comprehensive scenario, we differentiate between cases where the model: (1) knows the information and does not hallucinate ('factually correct'), (2) knows the information but hallucinates ('$\mathsf{HK}^+$'), and (3) does not know the information and thus hallucinates ('$\mathsf{HK}^-$'). In Figure 3 (Top), we see the detection accuracy results for the 3 classes across the model's layers using the bad-shot setting. The accuracy at the highest layer is 60%–70%, well above the random baseline of 33% (dashed line). Additionally, when we evaluate the accuracy of any two of the three classes produced by the classifier in the middle layer (16), we observe that the accuracy is no less than 70%. This high result indicates that models' inner states contain information for differentiating between the 3 cases. Figure 3 (Bottom) shows similar trends using the Alice-Bob setting, although the results are lower in this case, as may be expected given the subtlety of this setting.

## 4.2 GENERALIZATION OF WACK HALLUCINATIONS ACROSS HALLUCINATION SETTINGS

Next, we examine whether the Bad-shot and Alice-Bob synthetic settings are suitable for investigating $\mathsf{HK}^+$. To this end, we assess the generalization of hallucination detection classifiers based on these settings. In particular, we evaluate how well a classifier trained on examples from the bad-shot setting generalizes to examples obtained with the Alice-Bob setting (Section 2.2). This presents a significant challenge due to the inherent differences in the prompt between the bad-shots and the Alice-Bob-prompt (unlike the experiment in Section 4.1). In this experiment, we evaluate the ability

Table 3: Comparison of HK$^+$ and HK$^-$ mitigation results on Bad-shots/Alice settings.

| Model | Data Set | HK$^+$ | HK$^-$ |
|-------|----------|--------|--------|
| Gemma | triviaQA | **8.9** (0.0) / **15.0** (0.0) | 8.0 (7.4) / 9.0 (9.5) |
|  | Natural Questions | **13.2** (0.0) / **18.4** (0.0) | 5.6 (4.9) / 6.6 (6.6) |
| Llama | triviaQA | **17.5** (0.0) / **19.0** (0.1) | 7.3 (6.0) / 8.6 (9.4) |
|  | Natural Questions | **21.8** (0.0) / **24.3** (0.0) | 4.9 (4.1) / 5.9 (4.9) |
| Mistral | triviaQA | **48.7** (0.0) / **18.8** (0.0) | 7.0 (6.4) / 8.4 (9.0) |
|  | Natural Questions | **49.4** (0.0) / **20.6** (0.0) | 5.5 (3.5) / 5.5 (5.5) |

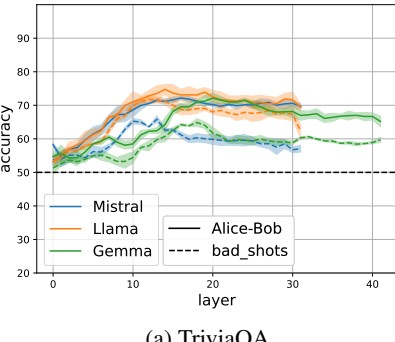

(a) TriviaQA.

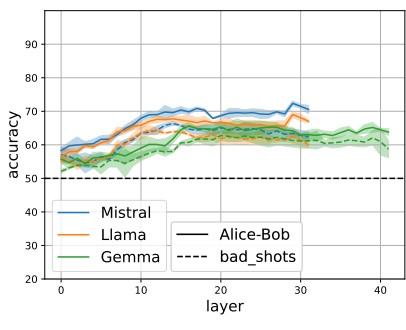

(b) Natural Questions.

Figure 4: Distinguishing factually correct from HK$^+$, when training on examples from either a bad-shot setting or an Alice-Bob setting, and testing on the Alice-Bob setting. While the change of setting reduces accuracy, the classifier still performs substantially above a random baseline.

to differentiate between HK$^+$ and factually-correct examples (binary classification), as the prompt settings are only aimed to split the knowledge examples into those two categories.

Figure 4 displays our findings. While changing the setting used for training the classifier reduces the results by up to 10%, it still performs above the random baseline. These results suggest some degree of generalization between different settings used to induce hallucinations despite knowledge. This conclusion lends support to the use of such synthetic datasets for studying the phenomenon of hallucinations despite knowledge. See Appendix E for generalization between Bad-shots configurations.

### 4.3 MITIGATION FAILS FOR HK$^-$ HALLUCINATIONS

Mitigating hallucinations is a broad area of research with approaches including prompting (Si et al., 2022; Dhuliawala et al., 2023), fine-tuning (Tian et al., 2023; Razumovskaia et al., 2023), and modifying model logits (Chuang et al., 2023; Kalai & Vempala, 2023; Li et al., 2023; Zhang et al., 2024). To demonstrate the necessity of distinguishing between HK$^-$ and HK$^+$ hallucinations, we show that each requires different mitigation techniques. In particular, HK$^-$ hallucinations, caused by a lack of knowledge, cannot be mitigated effectively through simple prompting alone; instead, they require an external knowledge source.

We applied the prompt "Generate answers that are entirely factual and precise, regardless of any issues in the text" at the start of each prompt. Table 3 shows the generation accuracy over the first 5 tokens for 1000 random examples of both HK$^-$ and HK$^+$ across Bad-shots and Alice-Bob settings. Each cell presents the mitigation results for the Bad-shots and Alice-Bob settings, with the pre-mitigation accuracy in parentheses. The values for HK$^+$ are significantly higher than those for HK$^-$, underscoring the importance of distinguishing between these hallucination types. Note that for HK$^-$, the values are not zero before mitigation (values in parentheses) and after, which we attribute to a wrong categorization of HK$^-$.

## 5 COMPARING MODEL-SPECIFIC AND GENERIC DATASETS

This section first compares WACK datasets crafted based on different models. Then it evaluates hallucination detection using model-specific vs. generic datasets.

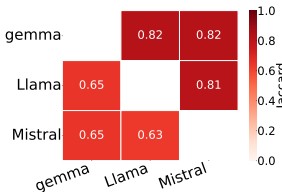 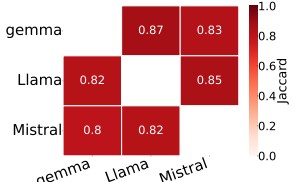 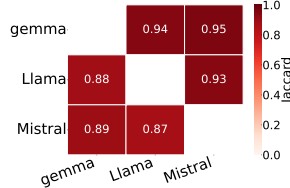

(a) Knowledge similarity between models.

(b) HK$^+$ similarity under Bad-shots-prompting.

(c) HK$^+$ similarity under Alice-Bob-prompting.

Figure 5: High-Knowledge and HK$^+$ differences on TriviaQA (above the diagonal) and Natural Questions (below the diagonal) between the models.

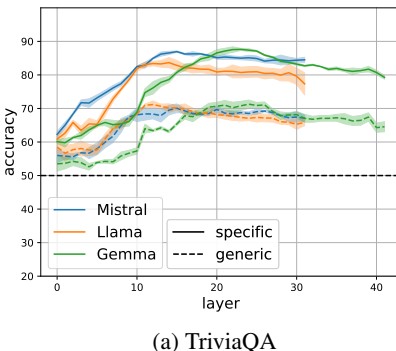 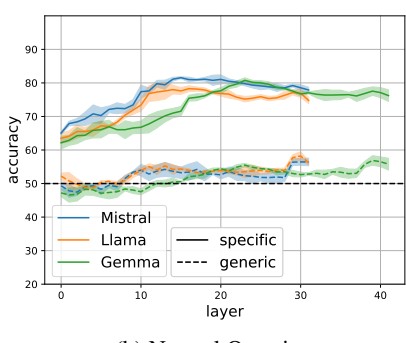

(a) TriviaQA

(b) Natural Questions

Figure 6: Distinguishing factually correct from HK$^+$ hallucinations using classifiers trained on generic vs. model-specific datasets.

## 5.1 DIFFERENT MODELS HAVE DIFFERENT KNOWLEDGE AND DIFFERENT HALLUCINATIONS

To demonstrate the heterogeneity in knowledge and hallucinations across models, we measure the Jaccard similarity (also known as intersection over union) of WACK datasets generated for different models. To compare knowledge of models, we calculate the Jaccard similarity of examples deemed as high-knowledge in two models, following our procedure from Section 2.1. To compare cases of hallucinations despite knowledge (HK$^+$), we calculate the Jaccard similarity of HK$^+$ examples in two models out of the set of examples that both models know.

Figure 5 displays these similarities. Jaccard values range from 0 (completely dissimilar) to 1 (perfect overlap). In Figure 5a, knowledge similarity for Natural Questions (below the diagonal) is approximately 0.6, indicating significant knowledge divergence between models. For TriviaQA (above the diagonal), models exhibit higher knowledge similarity (around 0.8).

Figures 5b and 5c reveal that hallucinations in shared knowledge cases are mostly similar (0.8–0.95). However, a 0.1–0.2 difference in similarity scores suggests each model still exhibits unique hallucination patterns. The bad-shot setting shows lower hallucination similarity than the Alice-Bob setting, indicating greater diversity in hallucination patterns for this scenario. These findings underscore the importance of model-specific approaches to hallucination detection and mitigation, as both knowledge bases and hallucination patterns vary across models and datasets.

## 5.2 DETECTING A MODEL'S HALLUCINATIONS IMPROVES WITH A MODEL-SPECIFIC DATASET

In this section, we show the importance of working with a model-specific dataset instead of a generic one. We start by explaining how to construct the generic dataset and then move to the experiments.

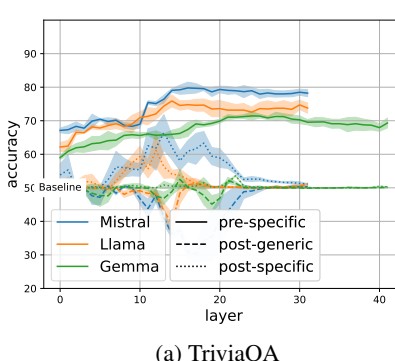 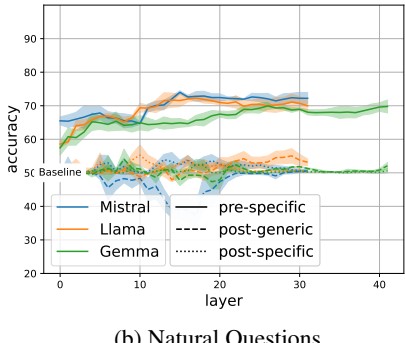

(a) TriviaQA                              (b) Natural Questions

Figure 7: Comparing $HK^+$ detection before generation using classifiers trained on model-specific pre-hallucination, generic post-hallucination, and model-specific post-hallucination examples.

**Generic Dataset.** A generic dataset is a labeled dataset that does not account for model-specific hallucinations or knowledge. Using a generic dataset is a common practice in the field of hallucination research for both detection and mitigation (e.g., Li et al. (2023); Chen et al. (2024); Zhang et al. (2024); Marks & Tegmark (2023); Hoscilowicz et al. (2024)). Thus, for comparison, we also compare to a generic dataset. The typical method for constructing a labeled QA-closed-book dataset involves using a triplet $q$, $a_g$, $a_h$, where $q$ is a question, $a_g$ is the gold answer, and $a_h$ is the hallucinated answer. A hallucination example is created by concatenating $a_h$ after $q$, while a factually-correct example is formed by appending $a_g$ after $q$. This labeling approach is based on the correctness of the answers relative to world knowledge. However, many datasets only include the $a_g$ answer, necessitating the creation of $a_h$. Following Li et al. (2023), we generate $a_h$ by prompting an LLM to produce a plausible yet incorrect answer. See Appendix B for dataset creation details.

**Results.** As this work focuses on cases of hallucinations despite knowledge ($HK^+$), we aim to show that the generic dataset is not effective in catching $HK^+$ hallucinations. Thus, we compare classifiers trained in two binary settings: (1) model-specific setting, separating $HK^+$ from factually correct examples; and (2) generic setting, separating hallucinations from factually correct examples. We test on the model-specific test set of $HK^+$ and factually correct examples. To make the generic and specific datasets more comparable, we add the bad-shots at the start of the prompt in both cases.

As Figure 6, shows classifiers trained on generic datasets (dashed lines) demonstrate varying degrees of effectiveness and are always worse than classifiers trained on model-specific datasets (solid lines). Notably, the model-specific classifiers maintain relatively high accuracy, unlike their generic counterparts. This comparison underscores the advantages of tailoring hallucination detection methods to individual models, suggesting that this approach more effectively captures model-specific nuances and leads to more reliable identification of hallucinations across various models. These results are consistent with related work (CH-Wang et al., 2023) that showed that a generic detector achieves less on specific datasets than training directly on specific datasets. Unlike us, they use a specific hallucination dataset that does not separate the two hallucinations.

### 5.3 PREEMPTIVE HALLUCINATION DETECTION USING MODEL-SPECIFIC DATASETS

Our previous detection results used hidden states obtained after the model generated an answer, potentially including a hallucination. A key advantage of model-specific datasets is their ability to detect potential hallucinations preemptively, *before they are generated*, a feature not possible with generic datasets. This section explores this capability using our WACK dataset (as before using $HK^+$ and factually correct examples), where each example contains only the question $q$ without an attached answer. As a result, the classifier is trained on the internal states of the examples at the last token of the question, rather than the last token of the answer. This approach allows us to analyze the model's propensity for hallucination based solely on the input query, a task unfeasible with generic datasets, which rely on concatenated answers for labeling, providing no signal of potential hallucination before generation.

Figure 7 displays preemptive hallucination detection results on the TriviaQA and Natural Questions datasets. Model-specific preemptive detection (solid lines) shows strong potential, indicating models can anticipate hallucinations. In contrast, generic post-hallucination detection (dashed lines) shows random (and even lower than random) performance, suggesting this approach is ineffective for identifying $HK^+$ hallucinations before they are generated. In comparison, model-specific hallucination detection after generation (dotted lines) yields varied outcomes: for the TriviaQA dataset, some layers and models achieve detection rates approaching 60%–70%, while for Natural Questions, the detection rates remain low and close to random. We conclude that post-hallucination settings are not effective for preemptive hallucination detection, further highlighting the benefits of model-specific datasets.

## 6 RELATED WORK

Our research investigates hallucination types ($HK^-$ and $HK^+$) and develops a methodology for constructing $HK^+$ hallucinations. It is related to research on hallucinations and jailbreaking.

**Hallucination Detection.** Detecting hallucinations can involve treating the model as a black box, posing questions or sampling its outputs (Gekhman et al., 2023; Pacchiardi et al., 2023; Manakul et al., 2023; Li et al., 2024a). Another line of work attempts to detect hallucinations, factuality, or answerability by examining the model's hidden representations, often by training a detection classifier (Burns et al., 2022; He et al., 2023; Rateike et al., 2023; Slobodkin et al., 2023; Azaria & Mitchell, 2023; CH-Wang et al., 2023; Yuksekgonul et al., 2023; Chen et al., 2023; Yin et al., 2024; Levinstein & Herrmann, 2024; Marks & Tegmark, 2023; Li et al., 2023). Most of this prior work used generic datasets. While we also employ detectors, we focus on model-specific datasets. Some prior work did explore model-specific hallucination datasets and showed their importance (Azaria & Mitchell, 2023; Ji et al., 2024; Cao et al., 2023; CH-Wang et al., 2023). However, these efforts did not differentiate between the causes of hallucinations ($HK^-$ and $HK^+$).

**Jailbreaking.** Jailbreaking refers to techniques for causing LLMs to generate unexpected or incorrect answers. For instance, Zhang et al. (2023) demonstrated the snowballing effect, where once the model outputs an it, it is more likely to generate an incorrect explanation for that fact. Additionally, research has shown that a model's answers can change due to persuasion, long conversations, fantasy settings, LLM personas, and out-of-distribution prompts (Zeng et al., 2024; Li et al., 2024b; Xu et al., 2023; Yao et al., 2023; Nardo, 2023; Joshi et al., 2023; Pacchiardi et al., 2023). These studies highlight that the correctness of a model's output depends on many characteristics of the prompt, allowing hallucinations to occur even when the model knows the correct answer.

While their work focuses on identifying methods that induce hallucinations, which can lead to $HK^+$ hallucinations, our investigation directly explores the $HK^+$ phenomenon and its relationship to $HK^-$. Additionally, we introduce a method to automatically construct the WACK dataset for further analysis.

## 7 DISCUSSION AND CONCLUSION

In this work, we emphasize the importance of differentiating between hallucinations caused by lack of knowledge ($HK^-$) and those occurring despite knowledge ($HK^+$). We introduced WACK, a method for creating model-specific datasets based on each model's knowledge and hallucinations. We proposed bad-shots and Alice-Bob settings to induce $HK^+$ hallucinations and showed some generalization between them. This indicates that these synthetic settings are effective for studying $HK^+$. Our findings reveal that each model has distinct knowledge and unique hallucination patterns, underscoring the need for model-specific datasets. Additionally, we demonstrated that generic datasets are less effective at detecting model-specific hallucinations than our tailored WACK datasets.

## 8 LIMITATIONS

Our work has a few limitations. While we evaluated three popular models, the patterns may differ in other ones. Additionally, we used only two settings to induce hallucinations given a model's correct knowledge; there may be many other ways to achieve similar aims. Finally, we only examined the two extremes of the knowledge spectrum, leaving the middle unexplored.

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

## A    HYPER PARAMETERS SEARCH FOR KNOWLEDGE CATEGORIZATION

Knowledge detection typically relies on the model's output, either through logits or generation. We focus on the generation approach, assessing whether the model consistently produces a factually correct answer among multiple samples, similar to recent work (Gekhman et al., 2024). This method is influenced by various hyperparameters including (1) number of generations, (2) sampling temperature, (3) length of generation, and (4) prompt structure.

As directly accessing factually correct is challenging, we instead examined the consistency of knowledge classification across different hyperparameter settings. A high similarity in categorization across settings would suggest comparable proximity to ground truth, reducing the impact of specific hyperparameter choices.

We evaluated the following hyperparameters:

- Shots: two different 3-shot examples and one zero-shot example.

- Temperature: $\{0.5, 1, 1.5\}$

- Number of generations: $\{5, 10\}$

- Length of generated text: $\{5, 10, 20\}$ tokens.

We started with a baseline configuration based on preliminary experiments: 3-shots, temperature of 0.5, 5 generations, and 5 tokens generated. We then modified one parameter at a time to assess its impact on classification similarity.

We categorized knowledge into three classes: "does not know" if the model did not generate the correct answer in any of the generations; "know" if the model always generated the answer; and "else" for anything in between.

We tested this approach on 1000 random TriviaQA examples across our three models. The average similarity among all 8 configurations (28 unique combinations) was 93.6% for Llama, 92.7% for Mistral, and 92.2% for Gemma, indicating a high consistency in knowledge classifications. The lowest similarity (about 80%) occurred with zero-shot configurations.

Based on these results, we adopted the baseline setting as our knowledge detector, using the 3-shot prompt corresponding to the bad-shots used in subsequent hallucination classification. The high similarity between different few-shot prompts suggests that varying the few-shot examples should yield comparable results. To enhance reliability, we supplemented this approach with one greedy generation, ensuring we capture the most likely output even if temperature-based generations fail to produce it.

Note that most examples are labeled as either "know" or "does not know" and only about 5% are labeled as "else". Thus we leave the treatment of this category for future work.

One area for improvement in future research is the method of answer detection. While we use 'exact match' for simplicity and achieve relatively good results, employing methods that allow for more flexible matching could enhance recall.

## B    DATASET CONSTRUCTION SPECIFICS

For TriviaQA we took 30K random examples from its training set as our initial dataset, making sure to use only examples where the answer was no longer than 5 tokens using the Mistral tokenizer. In addition, as we saw that some answers were written in upper case, we also used the lower-case version of these answers if they contained more than 3 letters and did not contain numbers or the '/' symbol.

For the Natural Questions datasaet, we also used 30K random examples, excluding examples with answer longer than 5 tokens as well as examples without an answer or with more than one answer. We again added lower-case versions of upper-case answers.

Table 4: Examples from the generic dataset of TriviaQA.

| Prompt | Factually correct | Hallucination |
|---|---|---|
| question: Which instrument was primarily played by band leader Count Basie? answer: | Piano | Trumpet |
| question: Into which body of water does the river Nile empty? answer: | Mediterranean Sea | Atlantic Ocean |
| question: Which planet has a 'great red spot'? answer: | Jupiter | Saturn |

### B.1 GENERIC DATASET CONSTRUCTION

To create the generic dataset, the key addition was obtaining an incorrect answer for each example. We generated these using Mistral (Jiang et al., 2023) with the following prompt:

```
Question: {question}
Correct Answer: {correct answer}
Incorrect Answer:
```

We accepted the model's greedy generation of 5 tokens as the incorrect answer if it did not contain the correct answer. During this process, we removed words such as 'Questions' or 'incorrect' that the model occasionally generated alongside the answer. For examples of hallucinated answers generated by Mistral in the generic dataset of TriviaQA, refer to Table 4.

Note that this process could potentially make the generic dataset more similar to the model-specific Mistral dataset. However, the main paper results show that this way of creating the wrong answers with Mistral provided no advantage for the generic dataset. In fact, this outcome further highlights the benefits of using a model-specific dataset. Despite the generic dataset being generated using responses from Mistral, it still performs worse than the model Mistral-specific dataset.

### B.2 SPECIFIC DATASET CONSTRUCTION

To construct a model-specific dataset, we began with all examples from the generic dataset. We then filtered these examples based on the model's knowledge, retaining only those classified as 'known' by the model (for detailed information on this classification process, refer to Appendix A).

Subsequently, we employed a prompt-based approach utilizing bad-shots to generate responses from the model. For each example, the model produced a greedy generation of 5 tokens. We then evaluated these generations to categorize the examples: if the correct answer was present within the generated tokens, we labeled the example as "factually correct"; otherwise, it was labeled as "hallucinated".

This process allowed us to create a dataset that specifically reflects the model's knowledge and generation tendencies, distinguishing between instances where the model correct responses and those where it generates hallucinated content.

Lastly, we provide the full list of the 20 bad shots and good shots we used:

**Good shots:**

question: What is the capital of France?
answer: Paris
question: How many continents are there?
answer: 7
question: Who wrote 'Romeo and Juliet'?
answer: William Shakespeare
question: What is the square root of 64?
answer: 8

question: Which element has the chemical symbol 'H'?
answer: Hydrogen
question: Who was the first President of the United States?
answer: George Washington
question: What is the powerhouse of the cell?
answer: Mitochondria
question: In what year did World War II end?
answer: 1945
question: What is the currency of Japan?
answer: Japanese Yen
question: Who painted the Mona Lisa?
answer: Leonardo da Vinci
question: What is the speed of light?
answer: 299,792 kilometers per second
question: How many sides does a hexagon have?
answer: 6
question: What is the boiling point of water in Celsius?
answer: 100 degrees
question: Who wrote 'To Kill a Mockingbird'?
answer: Harper Lee
question: What is the capital of Australia?
answer: Canberra
question: What is the largest ocean on Earth?
answer: Pacific Ocean
question: Who discovered penicillin?
answer: Alexander Fleming
question: What is the chemical symbol for gold?
answer: Au
question: What is the smallest prime number?
answer: 2
question: How many planets are there in our solar system?
answer: 8

**Bad shots:**

question: What is the capital of France?
answer: Berlin
question: How many continents are there?
answer: 6
question: Who wrote 'Romeo and Juliet'?
answer: Jane Austen
question: What is the square root of 64?
answer: 7
question: Which element has the chemical symbol 'H'?
answer: Helium
question: Who was the first President of the United States?
answer: Abraham Lincoln
question: What is the powerhouse of the cell?
answer: Golgi Apparatus
question: In what year did World War II end?
answer: 1939
question: What is the currency of Japan?
answer: Euro
question: Who painted the Mona Lisa?
answer: Pablo Picasso
question: What is the speed of light?
answer: 300,000 kilometers per second
question: How many sides does a hexagon have?
answer: 5

Table 5: Generated answers using greedy decoding on TriviaQA with Gemma.

| Type | Prompt | Golden | Generation | |
|---|---|---|---|---|
| | | | w/ without addition | w/ bad-shots |
| HK$^+$ | In the Old Testament, who was the mother of Solomon? | Bathsheba | Bathsheba | Mary |
| factually-correct | In humans, citguatera is an illness caused by eating contaminated what? | Fish | Fish | Fish |
| HK$^-$ | Who is the longest-serving captain in the history of Manchester United FC? | Bryan Robson | Roy Keane | Roy Keane |

question: What is the boiling point of water in Celsius?
answer: 50 degrees
question: Who wrote 'To Kill a Mockingbird'?
answer: J.K. Rowling
question: What is the capital of Australia?
answer: Sydney
question: What is the largest ocean on Earth?
answer: Atlantic Ocean
question: Who discovered penicillin?
answer: Isaac Newton
question: What is the chemical symbol for gold?
answer: Ag
question: What is the smallest prime number?
answer: 1
question: How many planets are there in our solar system?
answer: 9

### B.3 SPECIFIC DATASET QUALITATIVE EVALUATION

To illustrate the examples generated by our Bad-shot setting and to highlight the differences between factually-correct, HK$^+$, and HK$^-$ responses, we present sample generations with and without the Bad-shot setting in Tables 5, 6, and 7. We observe that for HK$^+$ examples, the model generates a hallucination only in the Bad-shots setting, while HK$^-$ examples produce hallucinations even without it. Additionally, factually-correct examples continue to generate the correct answer even in the Bad-shots setting.

## C ALICE-BOB-SETTING DATASET STATISTICS

We show in Table 8 the dataset statistics for the Alice-Bob setting. There are fewer hallucinations than in the bad-shot setting; however, there are still at least 600 hallucinations in each of the configurations, enabling sufficient investigation.

Table 6: Generated answers using greedy decoding on TriviaQA with Llama.

| | | | Generation | |
| Type | Prompt | Golden | w/ without addition | w/ bad-shots |
|---|---|---|---|---|
| HK$^+$ | Who in 1990 became the first chancellor of a united Germany | Helmut Kohl | Helmut Kohl | John F. Kennedy |
| factually-correct | In the human body, what is the more common name for the 'Zygomatic Bone'? | Cheekbone | Cheekbone | Cheekbone |
| HK$^-$ | In the song The Twelve Days of Christmas, how many pipers are there? | Eleven | 12 pipers | 12 |

Table 7: Generated answers using greedy decoding on TriviaQA with Mistral.

| | | | Generation | |
| Type | Prompt | Golden | w/ without addition | w/ bad-shots |
|---|---|---|---|---|
| HK$^+$ | What color are the stars on an official United States flag? | White | White | Red |
| factually-correct | A seriema is what type of creature? | Bird | A bird | A bird |
| HK$^-$ | Which insect does Isle of Wight Disease affect? | BEE | The Whitefly | Mosquito |

Table 8: Dataset label statistics on the Alice-Bob setting.

| Dataset | # Factually correct | # Hallucination (HK$^+$) | # Do-not-know (HK$^-$) |
|---|---|---|---|
| TriviaQA-Llama3-WACK | 14851 | 978 | 7356 |
| Natural-Questions-Llama3-WACK | 6059 | 979 | 14739 |
| TriviaQA-Gemma-WACK | 15418 | 679 | 6991 |
| Natural-Questions-Gemma-WACK | 7194 | 710 | 13762 |
| TriviaQA-Mistral-WACK | 14505 | 988 | 7650 |
| Natural-Questions-Msitral-WACK | 6232 | 876 | 14689 |

## D  DETECTION RESULTS ON THE MLP AND ATTENTION COMPONENTS

The results in the main paper are only shown using hidden states from the residual component of the LLMs, that is, the representations after each transformer block. To complete the picture, we provide detection results also for the MLP and attention components using the representations that are output by the component.

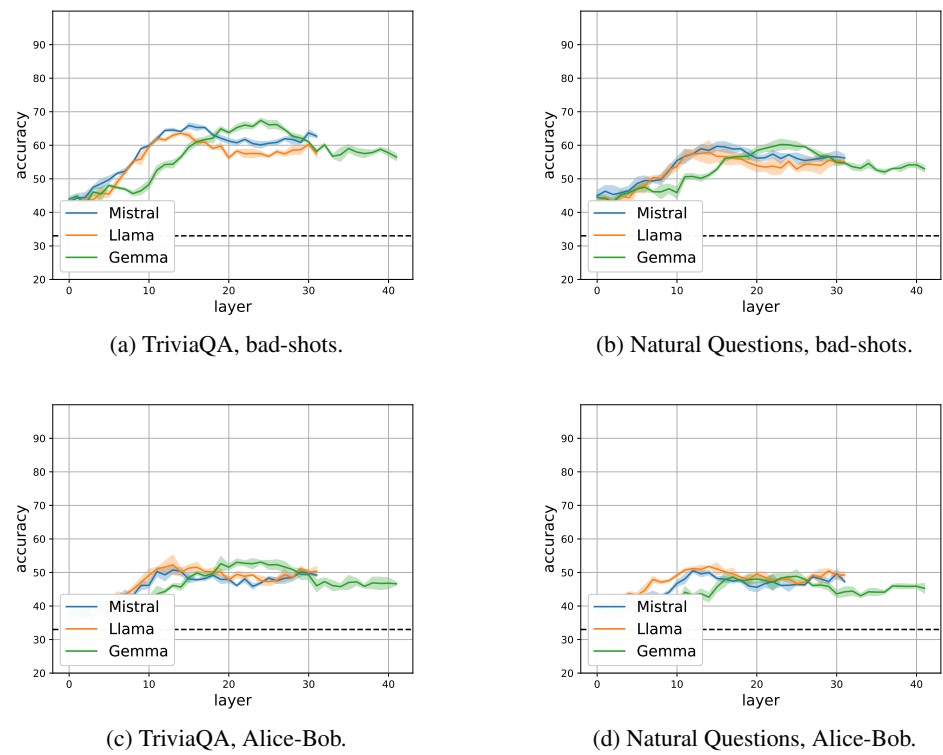

(a) TriviaQA, bad-shots.

(b) Natural Questions, bad-shots.

(c) TriviaQA, Alice-Bob.

(d) Natural Questions, Alice-Bob.

Figure 8: 3-way classification results into (i) hallucinations caused by lack of knowledge ($HK^-$), (ii) hallucinations caused despite having knowledge ($HK^+$), and (iii) factually correct examples. We show result using a bad-shot setting (Top) or an Alice-bob setting (Bottom) on MLP.

The results are shown in Figures 8 and 9 for the classification into the two hallucination types and factually correct examples, for the bad-shot and Alice-Bob settings.

Next in Figures 10 and 11 we see the results of the generalization of the bad-shot setting to the Alice-Bob setting using the MLP and Attention components.

Lastly, Figures 12 and 13 show detection of $HK^+$ hallucination results using classifiers trained on specific and generic datasets. Figures 14 and 15 give similar results when detecting using representations obtained before the hallucination occurs.

In all these figures, the results with the MLP and attention components yield similar trends to the ones in the main paper using the residual component, albeit with a moderately lower accuracy. This implies that the detection results are not limited to a specific component and are a broader phenomenon across components.

# E    GENERALIZATION BETWEEN BAD-SHOT SETTINGS

In the main paper, we used a single configuration with 3 bad shots, randomly selected. To demonstrate that our results are not dependent on this specific configuration, we conducted additional experiments with a different bad-shot setting. Specifically, we increased the number of shots to five and used a different random seed in the dataset creation process to sample different bad-shots. Our main objective was to show that the classifier trained under this new configuration performs similarly to the original 3-bad-shot setting. Demonstrating this would suggest that variations in bad-shot configurations do not impact classifier performance.

In Figure 16 we present the performance of a classifier trained in the 5-bad-shot configuration to distinguish between factually correct answers, $HK^-$, and $HK^+$. These results are comparable to those reported in the main paper in Section 4.1 for the Bad-shot and Alice-Bob configurations. This consistency indicates that the

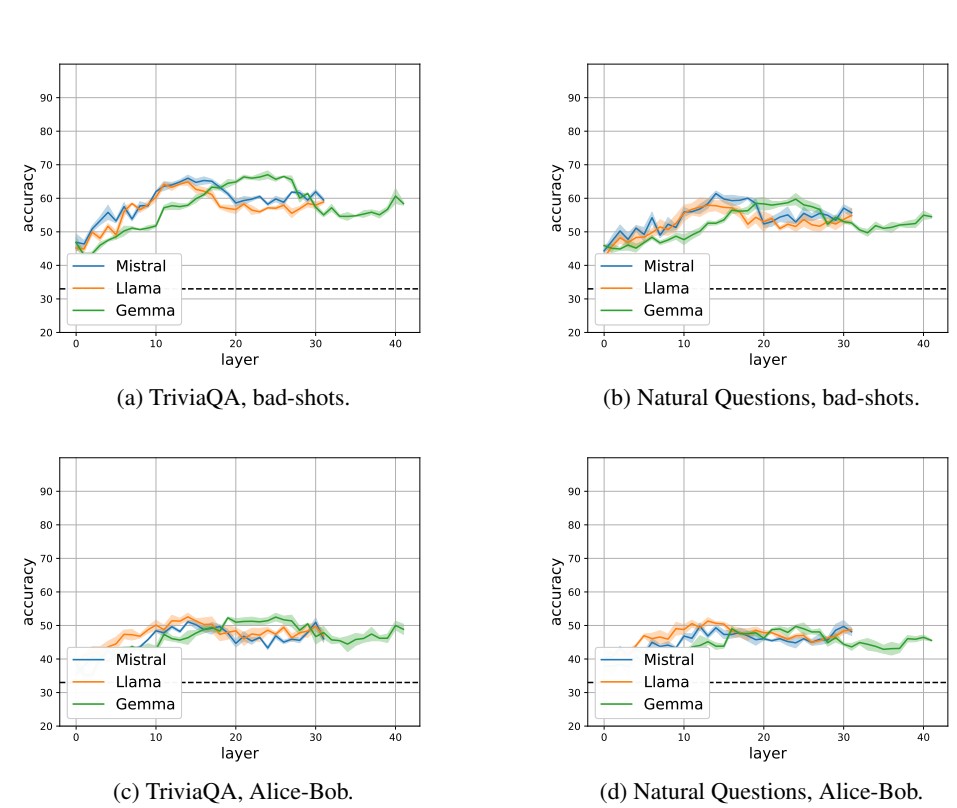

(a) TriviaQA, bad-shots.

(b) Natural Questions, bad-shots.

(c) TriviaQA, Alice-Bob.

(d) Natural Questions, Alice-Bob.

Figure 9: 3-way classification results into (i) hallucinations caused by lack of knowledge ($\mathsf{HK}^-$), (ii) hallucinations caused despite having knowledge ($\mathsf{HK}^+$), and (iii) factually correct examples. We show result using a bad-shot setting (Top) or an Alice-bob setting (Bottom) on Attention.

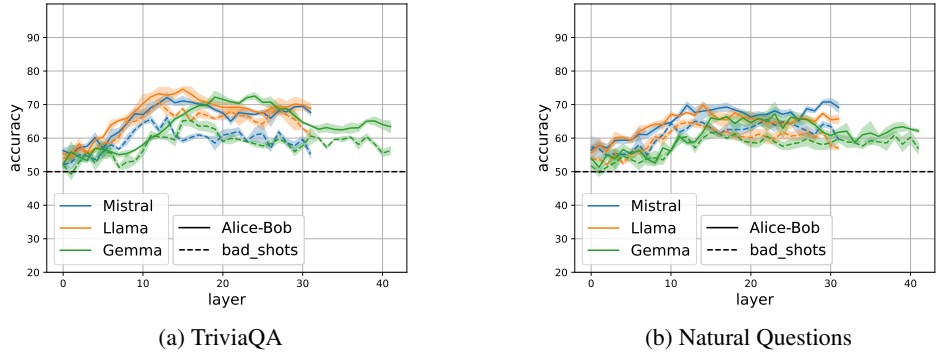

(a) TriviaQA

(b) Natural Questions

Figure 10: Distinguishing factually correct from hallucinations despite knowledge ($\mathsf{HK}^+$), when training on examples from either a bad-shot setting or an Alice-Bob setting, and testing on the Alice-Bob setting. While the change of setting reduces accuracy, the classifier still performs substantially above a random baseline on MLP.

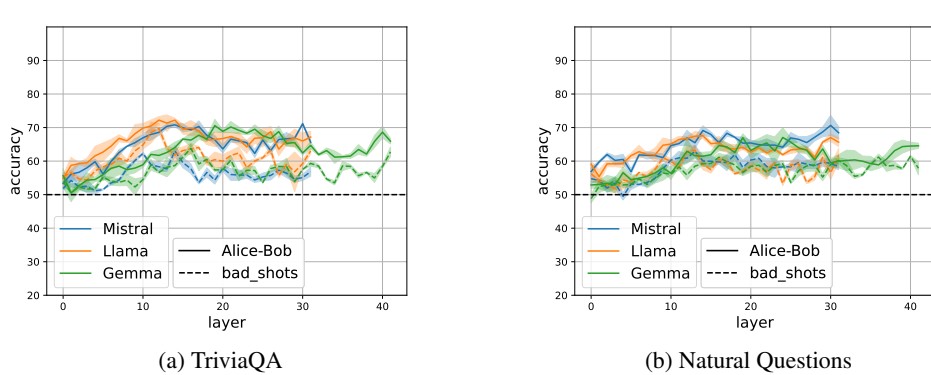

(a) TriviaQA

(b) Natural Questions

Figure 11: Distinguishing factually correct from hallucinations despite knowledge ($HK^+$), when training on examples from either a bad-shot setting or an Alice-Bob setting, and testing on the Alice-Bob setting. While the change of setting reduces accuracy, the classifier still performs substantially above a random baseline on Attention.

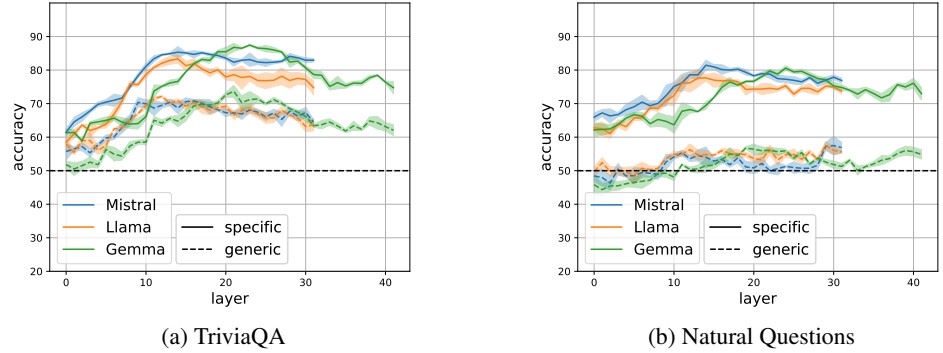

(a) TriviaQA

(b) Natural Questions

Figure 12: Distinguishing factually correct from $HK^+$ hallucinations using classifiers trained on generic vs. model-specific datasets on MLP.

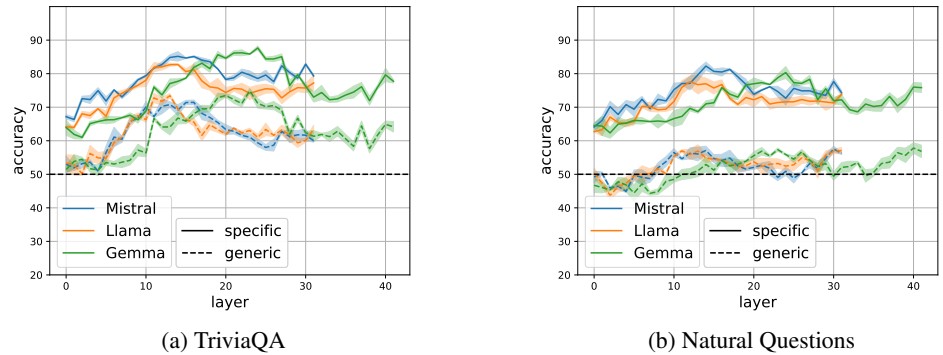

(a) TriviaQA

(b) Natural Questions

Figure 13: Distinguishing factually correct from $HK^+$ hallucinations using classifiers trained on generic vs. model-specific datasets on Attention.

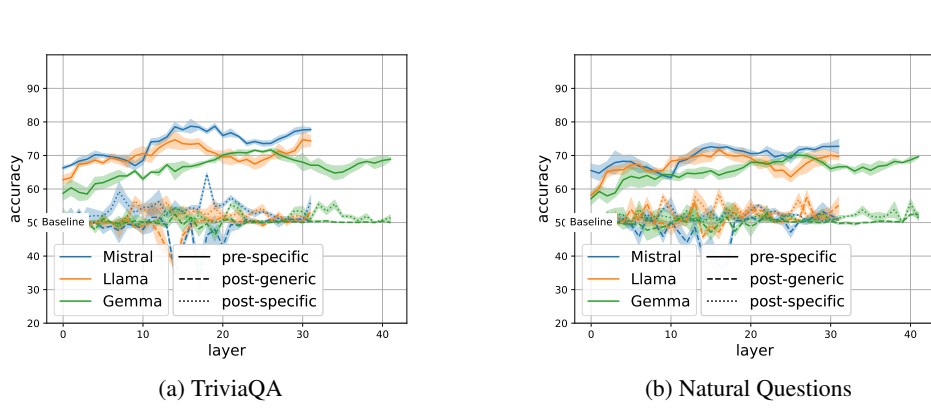

(a) TriviaQA  (b) Natural Questions

Figure 14: Comparing $HK^+$ detection before generation using classifiers trained on model-specific pre-hallucination, generic post-hallucination, and model-specific post-hallucination examples on MLP.

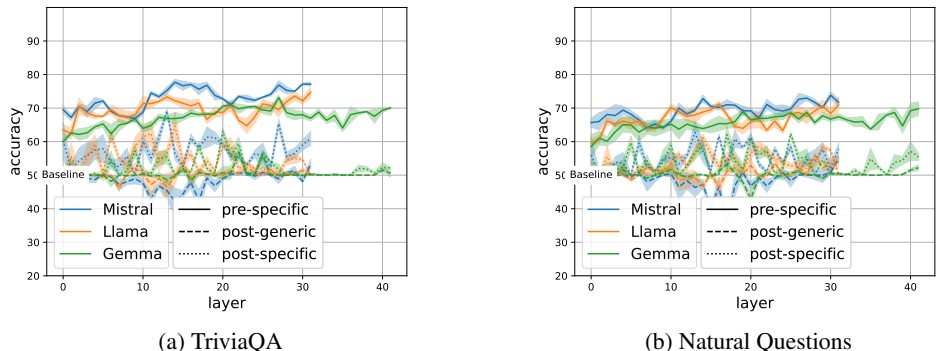

(a) TriviaQA  (b) Natural Questions

Figure 15: Comparing $HK^+$ detection before generation using classifiers trained on model-specific pre-hallucination, generic post-hallucination, and model-specific post-hallucination examples on Attention.

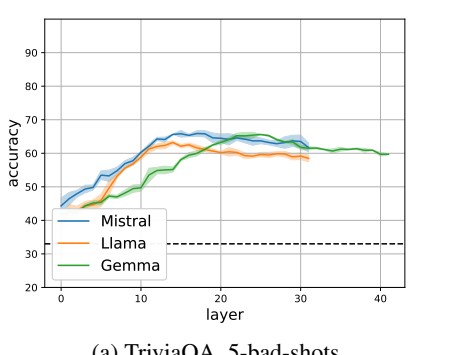 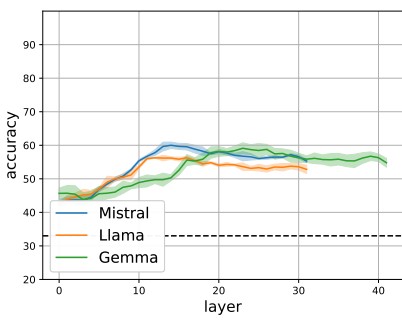

(a) TriviaQA, 5-bad-shots.

(b) Natural Questions, 5-bad-shots.

Figure 16: 3-way classification results into (i) hallucinations caused by lack of knowledge ($HK^-$), (ii) hallucinations caused despite having knowledge ($HK^+$), and (iii) factually correct examples. We show results using a bad-shot setting (Top) or an Alice-bob setting (Bottom).

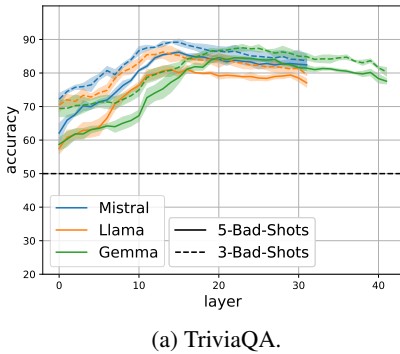 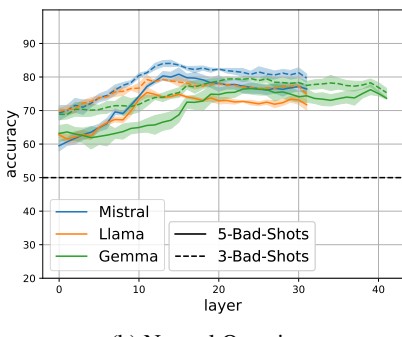

(a) TriviaQA.

(b) Natural Questions.

Figure 17: Distinguishing factually correct from hallucinations despite knowledge ($HK^+$), when training on examples from either a 3-bad-shot setting or a 5-bad-shot setting, and testing on the 5-bad-shot setting. The classifier performs substantially above a random baseline.

5-bad-shot configuration does not affect the classifier's ability to differentiate between the various types of hallucinations.

Additionally, Figure 17 shows the results of generalizing from the 3-bad-shot setting to the 5-bad-shot setting. The classifier in the 5-bad-shot setting achieved high performance in detecting $HK^+$, comparable to the results in the original 3-bad-shot configuration. Additionally, the 3-bad-shot setting generalized effectively to the 5-bad-shot configuration, indicating that variations in bad-shot configurations do not affect the classifier's detection abilities.

