# OpenReview forum: "Distinguishing Ignorance from Error in LLM Hallucinations"
_ICLR.cc/2025/Conference — ICLR 2025 Conference Withdrawn Submission_

### Official Review · Reviewer_YN6e · 2024-10-18

**Soundness:** 3
**Presentation:** 3
**Contribution:** 2
**Rating:** 5
**Confidence:** 5

**Summary:**

The paper addresses hallucinations in LLMs and introduces a new approach, Wrong Answer despite having Correct Knowledge (WACK), to differentiate between two types of hallucinations:

* HK-: Hallucinations due to lack of knowledge.
* HK+: Hallucinations despite the model having the required knowledge.

The paper constructs model-specific datasets to capture these types and demonstrates that such datasets outperform generic ones in detecting hallucinations. WACK probes internal states of the model to classify hallucinations more effectively across various prompt settings. This model-specific methodology generalizes between synthetic settings (Bad-shots and Alice-Bob) and allows better preemptive hallucination detection.

**Strengths:**

* This paper proposes a novel categorization and dataset, WACK, to study hallucinations based on knowledge availability.
* The experiments are well-designed and comprehensive, employing multiple models and tasks.
* The results highlight the importance of model-specific datasets and pave the way for future work on targeted mitigation of hallucinations.

**Weaknesses:**

* This paper primarily investigates two categories of reasons behind LLM hallucinations. However, the phenomenon of hallucination has been well studied and the contribution of this work is limited. The authors should further build on their findings to better alleviate hallucinations.
* The study is limited to three models in the 7B-9B range, raising questions about generalizability to larger models and commercial LLMs, e.g. gpt-series and claude.
* While the synthetic setups (Bad-shots and Alice-Bob) are innovative, real-world scenarios might involve more complex interactions, which are not captured.
* The focus on extreme knowledge scenarios (high- and low-knowledge) omits middle-ground cases, which could also be important.

**Questions:**

* How would the WACK approach extend to much larger LLMs like GPT-4 or PaLM?
* Is there any plan to incorporate more complex real-world prompts beyond the synthetic Bad-shot and Alice-Bob settings?
* How sensitive are the models to variations in the Bad-shot prompts? Could changing the bad-shots affect HK+ detection?

---

> ### Author Response · Authors · 2024-11-15
>
> We thank you for the thorough review and we are glad you recognized our novel categorization and datasets.
>
> Considering the comments, we have uploaded a revised version of our paper with modifications in blue. The main updates include adding mitigation results in Section 4.3, adding Appendix B.3 to provide additional qualitative evaluations, and adding Appendix E to present generalization and classification results across two different Bad-shot configurations.
>
> > This paper primarily investigates two categories of reasons behind LLM hallucinations. However, the phenomenon of hallucination has been well studied…
>
> Our paper’s primary aim is to demonstrate the existence of both HK- and HK+ hallucinations, a distinction that was not made in previous works. We propose a method to create model-specific labeled datasets and explore classifiers trained on internal states to differentiate between HK- and HK+.
>
> We agree that it can be beneficial to show that mitigation works better on HK+ than on HK- as HK- requires external knowledge for mitigation. There are many methods to mitigate hallucinations, from prompting (Si et al., 2022) to intervene in the model’s inner states (Chuang et al., 2023) We work with a simple form of mitigation via prompting. The final results are in the new paper version in a new Section 4.3. These results show that HK+ cases benefit much more from mitigation than HK-. Interestingly, a few HK- cases do benefit from mitigation, which we attribute to a wrong categorization as HK- in this case.
>
> > The study is limited to three models in the 7B-9B range, raising questions about generalizability to larger models and commercial LLMs...
>
> Commercial LLMs like GPT do not provide access to model activations, making them unsuitable for our research purposes. We emphasize that we have studied three of the strongest models with publicly available weights, which are widely used by the community. Thus we believe our conclusions are valuable to the large community using such models.
>
> > While the synthetic setups (Bad-shots and Alice-Bob) are innovative, real-world scenarios might involve more complex interactions, which are not captured.
>
> Our goal is to investigate HK+ hallucinations using two synthetic settings. Since these are synthetic, we needed to demonstrate that they can effectively mimic non-synthetic HK+ hallucinations, especially as we lack a large sample of non-synthetic HK+ cases. We showed that these two distinct settings are generalizable. This similarity supports the idea that different settings will produce similar HK+ classifiers, thus making those settings suitable for studying the HK+ phenomenon.
>
>
> > The focus on extreme knowledge scenarios (high- and low-knowledge) omits middle-ground cases, which could also be important.
>
> Our work distinguishes HK- hallucinations (low knowledge) from HK+ (high knowledge despite hallucination), omitting the middle ground where both factors may contribute.To focus on proving HK- and HK+ as distinct phenomena, we avoided complicating the discussion with the  middle-ground, which we plan to explore in future work. This is acknowledged in our limitations: “...we only examined the two extremes of the knowledge spectrum…”.
>
> > Is there any plan to incorporate more complex real-world prompts beyond the synthetic Bad-shot and Alice-Bob settings?
>
> We presented two settings: one more synthetic, the Bad-shots setting, and one more realistic, the Alice-Bob setting. In future work, we aim to explore additional settings. However, since our focus was to demonstrate distinct existence of HK+ and HK-, we chose not to focus on the various ways to create these hallucinations. This is acknowledged in our limitations: “...we used only two settings to induce hallucinations...”.
>
> > How sensitive are the models to variations in the Bad-shot prompts? Could changing the bad-shots affect HK+ detection?
>
> In Section 4.2, we evaluated generalization between the distinct Bad-shots and Alice-Bob settings. Despite their differences, both produced comparable HK+ detectors, showing that the classifiers are not overly specialized to a specific setting.
>
> To further address this, we conducted an additional experiment in which we modified both the number of shots (to 5) and used a different seed during dataset creation to generate different bad-shot sets. Our primary aim was to demonstrate that the classifier trained in this new configuration performs similarly to the original 3-bad-shot setting. This would indicate that variations in bad-shot permutations do not affect classifier detection of HK+. The results of this experiment are presented in Appendix E of the revised paper. We show that the generalization works well and that the new configuration achieves detection results comparable to those of the 3-bad-shot setting.
>
> We hope that we have addressed your questions adequately, and we are happy to continue the discussion.

---

> > ### Author Response · Authors · 2024-11-20
> >
> > Dear Reviewer,
> >
> > We hope our responses have addressed your concerns and would appreciate your reply to our response. We would be happy to engage further.

---

### Official Review · Reviewer_sVb3 · 2024-11-03

**Soundness:** 3
**Presentation:** 3
**Contribution:** 2
**Rating:** 5
**Confidence:** 3

**Summary:**

This paper proposes WACK, an approach for constructing model-specific datasets for hallucination detection with two types of hallucination samples. Leveraging existing closed-book question answering datasets, they first classify the samples into two categories based on whether the model can consistently answer the questions correctly in a closed-book setting. Then they prompt the model to generate hallucinated answers for those samples with correct predictions. They conduct analysis on the resulting dataset by training a linear probe on model’s inner states. Their experiments show that the proposed hallucination dataset is useful and better than a generic dataset for fine-grained model-specific hallucination detection.

**Strengths:**

- The presentation is good. The writing is well-structured and easy to follow.
- Hallucination detection in LLMs is important and timely. Compared with existing model-specific hallucination datasets, the proposed method can generate samples with a specific type of hallucination (i.e. the model is confused by the context despite having the relevant parametric knowledge), which enables more fine-grained hallucination detection.

**Weaknesses:**

- The motivation behind the prompting approach used in Alice-Bob setting is unclear.
- The synthetic hallucination dataset is model-specific and the data creation process is sensitive to the in-context hallucination samples in use. It is not clear how such datasets can be applied for evaluation in real-world scenarios since we cannot compare different models or different hallucination mitigation approaches on it.

**Questions:**

- Can you highlight the technical novelty in your work?  Is bad shots prompting and Alice-Bob prompting part of your technical contributions or did you adapt them from prior work?
- Did you measure the distribution similarity between the dataset generated by bad shots prompting and the one generated by Alice-Bob for the experiments in Section 4.2?
- I find the experiment design in Section 5.3 is confusing. Can you explain in more detail about how the probe is applied in this case and how is the setup different from the experiments in previous sections?

---

> ### Author Response · Authors · 2024-11-15
>
> Thank you for the thorough review, we are glad you recognized the importance of the topic and the benefit of our model’s specific dataset.
>
> Considering all the comments, we have uploaded a revised version of our paper with modifications highlighted in blue. The main updates include adding mitigation results in Section 4.3, creating Appendix B.3 to provide additional qualitative evaluations, and adding Appendix E to present generalization and classification results across two different Bad-shot configurations.
>
> > The motivation behind the prompting approach used in Alice-Bob setting is unclear.
>
> The Alice-Bob setting is a milder, more realistic approach to generating HK+ hallucinations compared to the Bad-shot setting. It does not contain bad-shots but a short story with small modifications, more similar to regular interactions with the LLM. We included both settings to demonstrate that these two distinct settings are generalizable, meaning that despite their differences, both approaches produce comparable HK+ examples in the model’s inner states. This similarity supports their suitability for studying and understanding the HK+ hallucination phenomenon.
>
> > The synthetic hallucination dataset is model-specific and the data creation process is sensitive to the in-context hallucination samples in use…
>
> In Section 5, we showed that a model-specific dataset is more effective at detecting the model's HK+ hallucinations than a generic dataset. While this approach introduces an additional overhead, we believe a model-specific dataset is essential for investigating this phenomenon and it is a key finding of our paper.
>
> Regarding sensitivity to in-context learning, our results in Section 4.2 demonstrate that the bad-shot setting generalizes well to the Alice-Bob setting. We showed that these two distinct settings are generalizable, indicating that, despite their large differences, both approaches yield comparable HK+ examples. This similarity supports the idea that different settings will produce similar HK+ classifiers, thus making those settings suitable for studying and understanding the HK+ hallucination phenomenon.
>
> In addition, the Alice-Bob setting is meant to show a more realistic setting without bad-shots but a simple story with small modifications that can induce the model to hallucinate.
>
> > Can you highlight the technical novelty in your work? Is bad shots prompting and Alice-Bob prompting part of your technical contributions or did you adapt them from prior work?
>
> Our paper’s primary aim is to demonstrate the existence of both HK- and HK+ hallucinations and to present a methodology for creating a model-specific, labelled dataset containing these examples to study them. We then investigate characteristics of those hallucinations using the model’s inner states. As part of our method we constructed the novel bad-shot and Alice-Bob settings to create HK+ hallucinations. See section 2.2 “To facilitate the creation of \wak hallucinations in large quantities, we design two synthetic setups that align with the ideas described above…”
>
> > Did you measure the distribution similarity between the dataset generated by bad shots prompting and the one generated by Alice-Bob for the experiments in Section 4.2?
>
> Alice-Bob and Bad-shot are distinct settings with significantly different proportions of HK+ and factually-correct examples, as shown in Tables 2 and 4. The similarity in labelling between factually-correct and HK+ examples ranges from 10% to 26% across all models and datasets. Thus, to demonstrate that these synthetic settings can be used to study HK+ hallucinations, we verified their generalizability. In Section 4.2, we ran a generalization test showing that a classifier trained on the Bad-shots setting can successfully detect HK+ examples in the Alice-Bob setting (a milder, more realistic setting).
>
> > I find the experiment design in Section 5.3 is confusing. Can you explain in more detail about how the probe is applied in this case…
>
> Preemptive detection refers to detecting hallucinations before the generation of the answer. In a typical text, the generated answer appears at the end. In this setting, the only modification is that we remove the answer and instead train the detection classifier on the representation after the model has processed the text before generating the answer.
>
> We clarified this in the paper by adding “…As a result, the classifier is trained on the internal states of the examples at the last token of the question, rather than the last token of the answer.”
>
> We hope that we have addressed your questions and concerns adequately, and we are happy to continue the discussion.

---

> > ### Author Response · Authors · 2024-11-20
> >
> > Dear Reviewer,
> >
> > We hope our responses have addressed your concerns and would appreciate your reply to our response. We would be happy to engage further.

---

### Official Review · Reviewer_1XKR · 2024-11-04

**Soundness:** 2
**Presentation:** 1
**Contribution:** 2
**Rating:** 5
**Confidence:** 4

**Summary:**

This paper categorizes LLM hallucination into two types: (1) cases where the LLM lacks sufficient information within its parameters to generate a correct answer (denoted as HK-), and (2) cases where the LLM provides an incorrect answer despite having the relevant information within its parameters (denoted as HK+). Focusing on the second type, this paper proposes a method for constructing model-specific datasets. The classifier trained with these datasets demonstrates the ability to classify HK+ types based on the model’s internal state.

**Strengths:**

- Dividing hallucination into two types can be useful, as suggested by the authors. Each hallucination type (HK+, HK-) can be mitigated using different methods (e.g., HK- can be addressed with external knowledge).
- Providing datasets that include a new type of hallucination (HK+) can benefit the NLP community.

**Weaknesses:**

If these below points can be clarified through additional responses from the authors, I would be glad to adjust my evaluation accordingly.

- The justification for categorizing hallucination types is weak. The paper does not sufficiently verify whether HK+ and HK- are truly mitigated by different methods (e.g., the performance improvement with external knowledge for HK- may be greater than for HK+). Additionally, the analysis supporting the claim that HK+ and HK- have distinct characteristics is inadequate. For instance, the experiment in Section 4.1 does not effectively demonstrate that the model’s internal state changes according to these hallucination types, contrary to what is asserted in line 311. The figures (e.g., Figure 3) display accuracy by model rather than by class (hallucination type). In summary, this paper should include stronger evidence to support the claim that distinguishing between HK+ and HK- is beneficial.

- The experimental detail is inconsistent. Figure 5 discusses "Knowledge and hallucination differences". Does this refer to 'factually correct' results, HK+, or HK-? If not, is it something new? Additionally, is there a distinction between the "probe" and the "classifier"? Furthermore, the meaning of the dashed line in Figure 3 is not explained (it appears to be a random baseline).

- In my opinion, some experimental setups and their significance are unclear. For instance, what is the meaning of the experiment in Section 4.2? In line 359, the authors mention that the Bad-shot and Alice-Bob synthetic settings are suitable for examining HK+, but it is unclear how this is demonstrated through generalization between the two settings. Additionally, in the experiment described in Section 5.2, the generic dataset should not account for model-specific hallucinations or knowledge. However, according to Appendix B.1, Mistral is used. Considering that Mistral is one of the three models used in the experiment, wouldn’t this also induce model-specific hallucinations and knowledge?

**Questions:**

- Is it possible to provide null accuracy for each classification result? (Figure 3,4,6,7)
- Was the classifier trained on different QA datasets (e.g., TriviaQA or NQ)? Training a new classifier for each model and each dataset could potentially be costly; what are the training costs and how many steps or training data is required? Could you provide experimental results related to training data efficiency?

---

> ### Author Response · Authors · 2024-11-15
>
> Thank you for the review. We are glad you recognize the benefit of distinguishing between HK+ and HK- and the benefit of the datasets.
>
> Considering the comments, we have uploaded a revised version of our paper with modifications in blue. The main updates include adding mitigation results in Section 4.3, creating Appendix B.3 to provide additional qualitative evaluations, and adding Appendix E to present generalization and classification results across two different Bad-shot configurations.
>
> > The justification for categorizing hallucination types is weak…
>
> Thanks for the suggestion. We agree that it can be beneficial to show that mitigation works better on HK+ than on HK- as HK- requires external knowledge for mitigation. There are many methods to mitigate hallucinations, from prompting (Si et al., 2022) to intervene in the model’s inner states (Chuang et al., 2023) We work with a simple form of mitigation via prompting. The final results are in the new paper version in a new Section 4.3. These results show that HK+ cases benefit much more from mitigation than HK-. Interestingly, a few HK- cases do benefit from mitigation, which we attribute to a wrong categorization as HK- in this case.
>
> > analysis supporting the claim that HK+ and HK- have distinct characteristics is inadequate…
>
> Figure 3 shows the accuracy of classifiers in distinguishing between Factually-correct, HK-, and HK+ examples. Each sub-figure presents results for all three models, and we observe that accuracy for each model is well above random chance. This indicates that the model's internal states can effectively distinguish between these types of hallucinations.
>
> > The experimental detail is inconsistent. Figure 5 discusses "Knowledge and hallucination differences"...
>
> Thank you the comment. Our method begins by splitting examples based on the model's level of knowledge. We label low-knowledge examples as HK-, while high-knowledge examples are divided into factually correct and HK+ based on model generation in the specific setting. In Figure 5.1, knowledge similarity refers to cases where the model has high-knowledge prior to the factually correct vs. HK+ split. The hallucination examples in Figures 5.b and 5.c correspond to HK+ cases. We changed the caption accordingly.
>
> > distinction between the "probe" and the "classifier"
>
> To avoid confusion, we now exclusively use the term ‘classifier’ in place of ‘probe’ throughout the paper.
>
> > Furthermore, the meaning of the dashed line in Figure 3 is not explained…
>
> We agree and have specified that it is the random baseline.
>
> > In my opinion, some experimental setups and their significance are unclear. For instance, what is the meaning of the experiment in Section 4.2?...
>
> Our aim is to enable investigation into the phenomenon of HK+ hallucinations. To study this, we used two synthetic settings: the Alice-Bob setting and the Bad-shot setting. Since these are synthetic, we needed to demonstrate that they can effectively mimic non-synthetic HK+ hallucinations, especially as we lack a large sample of non-synthetic HK+ cases. We showed that these two distinct settings are generalizable, indicating that, despite their large differences, both approaches yield comparable HK+ examples. This similarity supports the idea that different settings will produce similar HK+ classifiers, thus making those settings suitable for studying and understanding the HK+ hallucination phenomenon.
>
> > Additionally, in the experiment described in Section 5.2, the generic dataset should not account for model-specific hallucinations or knowledge. However…
>
> To construct the generic dataset, we generated incorrect answers as generic hallucinations by prompting Mistral to produce wrong answers. This process could potentially make the generic dataset more similar to the model-specific Mistral dataset. However, our results show that this way of creating the wrong answers with Mistral provided no advantage for the generic dataset. In fact, this outcome further highlights the benefits of using a model-specific dataset. Despite the generic dataset being generated using responses from Mistral, it still performs worse than the model Mistral-specific dataset. We added clarification in the new version.
>
> > Is it possible to provide null accuracy for each classification result? (Figure 3,4,6,7)
>
> The dashed line in all the graphs is the null-random accuracy. We added specification.
>
> > Was the classifier trained on different QA datasets...
>
> For each model and dataset, we train a separate classifier for each layer. Training takes only a few minutes per model and dataset. We use the LinearSVC classifier from the sklearn library, set with a maximum of 1 million iterations. As outlined in Section 3, we use 1,000 examples for each category (HK-, HK+, and factually correct), with 70% of the data for training and 30% for testing.
>
> We hope that we have addressed your questions and concerns adequately, and we are happy to continue the discussion.

---

> > ### Author Response · Authors · 2024-11-20
> >
> > Dear Reviewer,
> >
> > We hope our responses have addressed your concerns and would appreciate your reply to our response. We would be happy to engage further.

---

> > > ### Comment · Reviewer_1XKR · 2024-11-27
> > >
> > > Thank you for your response, and since some of my concerns have been addressed, I'm raising my score accordingly. However, I still have some concerns about a few details.
> > >
> > > As the authors mentioned in lines 371–372, in Table 3, the value of HK- is not zero before mitigation. Could you analyze the performance improvement of HK- by focusing only on the examples that were incorrect before mitigation?
> > >
> > > Additionally, to demonstrate that HK+ and HK- have distinct characteristics, I believe Figure 3 should display accuracy by class (hallucination type). I do understand that time may be limited, but would it be possible to show class-wise performance at least for Llama among the three models?

---

> > > > ### Author Response · Authors · 2024-11-27
> > > >
> > > > Thank you for raising the score.
> > > >
> > > > After removing all examples from HK- that were already incorrect before mitigation, the improvement due to mitigation is only 2-4% across all datasets, models, and settings. This low improvement, especially when compared to the higher results under HK+ (as shown in Table 3), highlights the need to address these two types of hallucinations separately and use distinct mitigation strategies for each.
> > > >
> > > > Due to time constraints, we only evaluated the proportional accuracy of each label (HK-, HK+, and factually correct) on Llama using TriviaQA under the Bad-shot setting. The accuracy reached 75% for factually correct, 64% for HK+, and 53% for HK-, compared to a baseline of 33%.
> > > >
> > > > We hope this addresses your concerns.

---

### Official Review · Reviewer_Z8gi · 2024-11-04

**Soundness:** 2
**Presentation:** 2
**Contribution:** 2
**Rating:** 3
**Confidence:** 4

**Summary:**

The study aims to detect two types of hallucinations in LLMs, (1) the model does not hold the correct answer in its parameters; (2) the model answers incorrectly despite having the required knowledge. To this end,  the study introduce Wrong Answer despite having Correct Knowledge (WACK), an approach for constructing model-specific datasets for the second hallucination type. Experimental results show that training a probe on the WACK datasets leads to better hallucination detection of case (2) hallucinations than using the common generic one-size-fits-all datasets.

**Strengths:**

1. This study intriguingly explores the resilience of knowledge in large language models (LLMs) when faced with incorrect demonstrations.
2. The findings indicate that a probing based on the proposed WACK dataset is successful in identifying knowledge within LLMs that is susceptible to being misled by incorrect demonstrations.

**Weaknesses:**

1.  What is the value to analyze the types of hallucinations, namely HK- (hallucination caused by lack of knowledge) and HK+ (hallucination despite knowledge)? The scenario involving HK+ seems impractical; in real-world situations, users are unlikely to provide numerous incorrect inputs to elicit the correct answer. A scenario where incorrect information subtly influences the dialogue might be more realistic.
2. The approach to detecting factual accuracy and HK+ is strange. If a model is prompted with incorrect examples, why should it be expected to produce the correct response? It appears that the study is intended to test the resilience or robustness of the model's internal knowledge against incorrect prompts, rather than focusing on the hallucination of internal knowledge itself.
3. Given the study aims to assess the robustness of knowledge in various contexts, the primary focus should be on the impact of bad interventions. The author should undertake a thorough and detailed analysis that includes the effects of varying numbers of demonstrations, the reasons for the induced answers, and so forth.
4. Why only use three negative examples for the study? The selection and impact of these hyper-parameters, such as the number and content of these negative cases, should be discussed. Additionally, it might be beneficial to compare these with simple templates that lack substantive information.
5. The paper could benefit from more precise writing. For instance, lines 115-118 should clearly define what is meant by "high-knowledge" and "low-knowledge." The current wording is vague and difficult to comprehend.
6. The section on related work discussing 'Jailbreak' does not seem relevant to this study. A discussion on the robustness of knowledge within large language models should be included to provide a more informative background.

**Questions:**

See weakness.

---

> ### Author Response · Authors · 2024-11-15
>
> Thank you for the review and for acknowledging our probes detection of hallucination despite knowledge (HK+) results.
> Below, we provide detailed answers to your questions and address each of your concerns.
>
> Considering all the comments, we have uploaded a revised version of our paper with modifications highlighted in blue. The main updates include adding mitigation results in Section 4.3, creating Appendix B.3 to provide additional qualitative evaluations, and adding Appendix E to present generalization and classification results across two different Bad-shot configurations.
>
> Based on some of your questions, it seems that the aim of our paper was not clear.
> Our paper’s primary aim is to demonstrate the existence of both HK- and HK+ hallucinations and to present a methodology for creating a model-specific, labeled dataset containing these examples. We then investigate the characteristics of these hallucinations using the model’s inner states. Our focus is not on assessing the resilience of the model's internal knowledge against incorrect prompts. Instead, we leverage this phenomenon of jailbreaking (Zeng et al., 2024; Li et al., 2024b; Xu et al., 2023; Yao et al., 2023; Nardo, 2023; Joshi et al., 2023; Pacchiardi et al., 2023) to create HK+ examples and evaluate the model’s ability to distinguish between HK+ and HK-. Furthermore, we show through the Alice-Bob setting that even a milder, more realistic approach of a short story with small modifications can lead to HK+ hallucinations.
>
> > Why only use three negative examples for the study? The selection and impact of these hyper-parameters, such as the number and content of these negative cases, should be discussed. Additionally, it might be beneficial to compare these with simple templates that lack substantive information.
>
> We used three bad-shots rather than a lower number because fewer Bad-shots yield significantly fewer HK+ examples; for instance, 1-Bad-shot produces only 20–40% as many HK+ hallucinations.  And we needed a big enough number of HK+ examples for our evaluation. Additionally, we chose not to use more Bad-shots than necessary, as our goal was to create a setting with only a limited number of mistakes making it more realistic yet able to yield enough HK+ examples.
>
> To further address this, we conducted an additional experiment in which we modified both the number of shots (increased from 3 to 5) and used a different seed during dataset creation to generate different bad-shots sets. Our primary aim was to demonstrate that the classifier trained in this new configuration performs similarly to the original 3-bad-shot setting. If successful, this would indicate that variations in bad-shot permutations do not affect classifier performance.
> The results of this experiment are presented in Appendix E of the revised paper. We show that the generalization works well and that the 5-bad-shot configuration achieves detection results comparable to those of the 3-bad-shot setting.
> We also demonstrated in the paper that even a distinct setting, such as the Alice-Bob setting, can be detected using a classifier trained on bad-shot examples. This shows that, despite differences between settings, HK+ examples share some similarities in their internal states, making them detectable across settings.
>
>
> > The paper could benefit from more precise writing. For instance, lines 115-118 should clearly define what is meant by "high-knowledge" and "low-knowledge." The current wording is vague and difficult to comprehend.
>
> In section 2.1 we explain that knowledge is a spectrum in terms of what are considered high-knowledge and low-knowledge cases, and how to create examples for each.
> To clarify we modified line 112 to:
> ”We refer to the model's parametric knowledge as being at the 'low-knowledge end' when there is little to no association between a_g​ and q, and as the 'high-knowledge end' when this association is strong.”
> In the following paragraph, we discuss the specificity of how to classify examples to low-knowledge and high-knowledge cases.
>
> > The section on related work discussing 'Jailbreak' does not seem relevant to this study. A discussion on the robustness of knowledge within large language models...
>
> The primary goal is to demonstrate the existence of both HK- and HK+ hallucinations and to present a methodology for creating a model-specific, labeled dataset containing these examples. To achieve this, we developed two settings inspired by ideas from jailbreak papers.
> Our aim is to show that hallucinations can occur despite the model’s knowledge. We do not focus on evaluating the robustness of knowledge but focus on simple settings (one more realistic and one less realistic) that elicit HK+ hallucinations, enabling us to study these phenomena more effectively.
>
> We hope that we have addressed your questions and concerns adequately, and we are happy to continue the discussion.

---

> > ### Author Response · Authors · 2024-11-20
> >
> > Dear Reviewer,
> >
> > We hope our responses have addressed your concerns and would appreciate your reply to our response. We would be happy to engage further.

---

> ### Author Response · Authors · 2024-11-26
>
> The HK+ phenomenon occurs when a model generates incorrect answers despite knowing the correct ones. This can happen in various situations, such as the snowballing effect [Zhang et al., 2023], persuasive styles [Xu et al., 2023; Zeng et al., 2024], or even due to irrelevant text [Shi et al., 2023], all of which influence the model's hallucinations and can happen organically.
>
> Beyond the Bad-shot setting, we also use the Alice-Bob setting, a milder and more realistic scenario for inducing HK+ hallucinations. Unlike the Bad-shot setting, it does not include factually incorrect examples, yet it still produces a large number of hallucinations (a few percentages out of all the model knowledge examples). These setups allow us to study HK+ hallucinations in controlled conditions without requiring lengthy conversations.
>
> In the Bad-shot setting, the model is prompted with three incorrect examples ("bad shots") where the answers are factually wrong. Notably, just three such examples significantly increase the rate of hallucinations. These hallucinations can stem from the model's prior mistakes or the user's lack of knowledge. As shown in [Zhang et al., 2023], a single hallucination can trigger a snowballing effect: if the model produces one error, it is more likely to justify it with further incorrect explanations. Thus, the hallucination might be influenced by the model’s earlier errors during the conversation.
>
> When the prompt is reduced to a one-Bad-shot setting, the number of hallucinations drops to 20–40% of those seen in the three-Bad-shot setup. This is still a substantial number of hallucinations —especially given the short prompt. To better study HK+ hallucinations, we chose a setup with more frequent hallucinations, though milder settings also exhibit this phenomenon.
>
> Hope this addressed your concerns adequately, we are happy to continue the discussion.

---

### Official Review · Reviewer_7f6D · 2024-11-04

**Soundness:** 3
**Presentation:** 3
**Contribution:** 3
**Rating:** 6
**Confidence:** 4

**Summary:**

The paper proposes a method to separate two causes for LLM hallucination – when the model does not have the related parametric knowledge, and when the model lies despite of having the required knowledge.

**Strengths:**

-Most of existing methods tend to misuse the term of hallucination and mix up these two cases. So this paper is potentially impactful to open up a new direction.
-The paper proposes an effective framework to generate good-shot and bad-shot examples.

**Weaknesses:**

-There are quite a few unexplained and confusing terms, such as "snowball effect" and "high knowledge".
-It would be great if the authors can add a discussion section about how to leverage this categorization for hallucination mitigation.
-The experiment section lacks of qualitative analysis to make the results more insightful.
-Section 5.3 preemptive detection of hallucination is an interesting idea, but the description is very vague and would be difficult to duplicate.

**Questions:**

-In the bad-shot setting, how important is it to generate diverse examples? If it's crucial, how did you make sure the three random examples are representative?

---

> ### Author Response · Authors · 2024-11-15
>
> Thank you for the thorough review and your valuable comments. We are glad you recognized the potential impact of our work.
>
> Considering all the comments, we have uploaded a revised version of our paper with modifications highlighted in blue. The main updates include adding mitigation results in Section 4.3, creating Appendix B.3 to provide additional qualitative evaluations, and adding Appendix E to present generalization and classification results across two different bad-shot configurations.
>
> > There are quite a few unexplained and confusing terms, such as "snowball effect" and "high knowledge"
>
> In line 131 we explain that “... demonstrated that after a model produced an incorrect answer, it was likely to generate an incorrect explanation to justify its error, which they termed the “snowballing effect””
>
> In section 2.1 we explain that knowledge is a spectrum in terms of what are considered high-knowledge and low-knowledge cases, and how to create examples for each.
> To clarify we modified line 112 to:
> ”We refer to the model's parametric knowledge as being at the 'low-knowledge end' when there is little to no association between a_g​ and q, and as the 'high-knowledge end' when this association is strong.”
> In the following paragraph, we discuss the specificity of how to classify examples to low-knowledge and high-knowledge cases.
>
>
> > It would be great if the authors can add a discussion section about how to leverage this categorization for hallucination mitigation.
>
> Thanks for the suggestion. We agree that it can be beneficial to show that mitigation works better on HK+ than on HK- as HK- requires external knowledge for mitigation. There are many methods to mitigate hallucinations, from prompting (Si et al., 2022) to intervene in the model’s inner states (Chuang et al., 2023) We work with a simple form of mitigation via prompting. The final results are in the new paper version in a new Section 4.3. These results show that HK+ cases benefit much more from mitigation than HK-. Interestingly, a few HK- cases do benefit from mitigation, which we attribute to a wrong categorization as HK- in this case.
>
> > The experiment section lacks of qualitative analysis to make the results more insightful.
>
> Thanks for the comment. In Section 2.2, we provide qualitative examples of the model’s generations with good-shots and bad-shots, illustrating how the outputs change in the bad-shot setting, we can see that in the good shot setting the model output the correct answer and in the bad-shot setting it outputs a hallucination. Additionally, we added in the new version  Appendix B.3.  We present there examples for each model, including HK+, HK-, and factually correct generations. We can see there that for the factually-correct examples the model generates the correct answer with and without the bad-shot settings, for HK+ it generates the correct answer without the bad-shot setting and with it a hallucination, and lastly for HK- it generates a hallucination with and without the bad-shots.
>
> > Section 5.3 preemptive detection of hallucination is an interesting idea, but the description is very vague and would be difficult to duplicate.
>
> Preemptive detection refers to detecting hallucinations before the generation of the answer. In a typical text, the generated answer appears at the end. In this setting, the only modification is that we remove the answer and instead train the detection classifier on the representation after the model has processed the text before generating the answer.
>
> We clarified this in the paper by making the following modification: ”This section explores this capability using our WACK dataset (as before using HK+ and factually correct examples), where each example contains only the question q without an attached answer. As a result, the classifier is trained on the internal states of the examples at the last token of the question, rather than the last token of the answer.”
>
>
> > In the bad-shot setting, how important is it to generate diverse examples? If it's crucial, how did you make sure the three random examples are representative?
>
> We agree that evaluating this aspect is important. To address this, we conducted an additional experiment in which we modified both the number of shots and used a different seed during dataset creation to generate different bad-shot sets. Our primary aim was to demonstrate that the classifier trained in this new configuration performs similarly to the original 3-bad-shot setting. If successful, this would indicate that variations in bad-shot permutations do not affect classifier detection of HK+.The results of this experiment are presented in Appendix E of the revised paper. We show that the generalization works well and that the 5-bad-shot configuration achieves detection results comparable to those of the 3-bad-shot setting.
>
> We hope that we have addressed your questions and concerns adequately, and we are happy to continue the discussion.

---

> > ### Author Response · Authors · 2024-11-20
> >
> > Dear Reviewer,
> >
> > We hope our responses have addressed your concerns and would appreciate your reply to our response. We would be happy to engage further.

---

### Note · Authors · 2024-12-11

I have read and agree with the venue's withdrawal policy on behalf of myself and my co-authors.